# Internet Addiction and Polish Women’s Sexual Functioning: The Role of Social Media, Online Pornography, and Game Use during the COVID-19 Pandemic—Online Surveys Based on FSFI and BSMAS Questionnaires

**DOI:** 10.3390/ijerph19138193

**Published:** 2022-07-04

**Authors:** Anna Pawlikowska, Ewa Szuster, Paulina Kostrzewska, Amanda Mandera, Małgorzata Biernikiewicz, Małgorzata Sobieszczańska, Krystyna Rożek-Piechura, Monika Markiewicz, Agnieszka Rusiecka, Dariusz Kałka

**Affiliations:** 1Cardiosexology Students Club, Wroclaw Medical University, 50-368 Wroclaw, Poland; anna.pawlikowska96@gmail.com (A.P.); pkostrzewska1@gmail.com (P.K.); amanda.mandera@vp.pl (A.M.); 2Studio Słowa, 50-357 Wroclaw, Poland; malgorzata@biernikiewicz.pl; 3Clinical Department of Geriatrics, Wroclaw Medical University, 50-369 Wroclaw, Poland; malsobie100@gmail.com; 4Faculty of Physiotherapy, Wroclaw University of Health and Sport Sciences, 51-612 Wroclaw, Poland; krystyna.rozek-piechura@awf.wroc.pl (K.R.-P.); dariusz.kalka@gmail.com (D.K.); 5Men’s Health Centre in Wrocław, 53-151 Wroclaw, Poland; tomczykmonik@gmail.com; 6Statistical Analysis Centre, Wroclaw Medical University, 50-367 Wroclaw, Poland; agnieszka.rusiecka@umw.edu.pl

**Keywords:** social media, FSFI, BSMAS, sexual activity, female

## Abstract

In this study, we investigated the relationship between social media use and women’s sexual functioning during the COVID-19 pandemic. Data were collected between April and November 2021. Online surveys including the Female Sexual Functioning Index (FSFI) and Bergen Social Media Addiction Scale (BSMAS) questionnaires were distributed to young, sexually active women. Information was collected on their demographics, sexual life, and use of social media. We enrolled 546 women (mean age 23.07 ± 4.69). In general, 5.68% of the women were at high risk of social media addiction. Social media addiction had a negative impact on FSFI scores, while pornography use had a positive effect on women’s sexual functioning. Users of dating apps also obtained lower FSFI scores than non-users. No differences in FSFI scores were observed between gamers and non-gamers. The impact of time spent on social media on FSFI scores was not significant. We conclude that social media addiction negatively affected women’s sexual functioning during the COVID-19 pandemic.

## 1. Introduction

Today, the Internet is an integral part of everyday life. As of April 2022, there were 5 billion internet users worldwide, representing 63% of the global population. Of this total, 4.65 billion were users of social media [1]. It is estimated that young adults spend up to twelve hours a day on social media (Vaterlaus, J. M., Patten, E. V., Roche, C., & Young, J. A. (2015). #Getting Healthy: The perceived influence of social media on young adult health behaviors. Computers in Human Behavior, 45, 151–157) [2]. Furthermore, the percentage of adults in the US who use social media increased from 5% in 2005 to 79% in 2019. The most commonly used platforms are Facebook (2.38 billion users) and YouTube, then, in decreasing order of users, WhatsApp, WeChat, Instagram, TikTok, Weibo, Reddit, Twitter, Pinterest, and Snapchat. However, the popularity of social media platforms differs by age. This rule particularly applies to Instagram and Snapchat, for which the age gradient is very steep [2].

Similarly, the number of gamers rises every year. FinancesOnline reports that there were 2.69 billion gamers in the world by the end of 2020 [3]. In 2021, women represented 45% of gamers in the US [4]. In Poland, 87% of people have access to the Internet. People aged 16 to 64 years spend about 6 h and 39 min online daily, including 1 h and 49 min on social media [5].

The digital era has a significant impact on every area of life. In recent years, the way people find love has also diametrically changed, and online dating has become increasingly popular.

By 2024, it is forecast that there will be nearly 280 million online users of dating services, including 113 million users of matchmaking services and 70 million casual daters [6]. The most popular dating website in Poland is badoo.pl, while the most often downloaded is Tinder [7]. Summarizing the numbers mentioned above, it is easy to observe that digital media and the Internet can also have a negative impact on human health. In 1996, Young introduced the concept of Internet addiction disorder [8]. Furthermore, National Family Week reported that 40% of girls aged 8–15 years identified Facebook as one of the most important things in their lives, compared to 6% of boys [9]. Moreover, watching pornography online is also a part of online time. Every second, 28,258 users are watching pornography on the Internet [10]. Rozgonjuk et al. concluded that the negative impact of social media on everyday life and efficiency at work were significantly correlated with the intensity of all disorders related to the use of social networks [11].

In addition, the COVID-19 pandemic has changed people’s everyday lives in numerous fields. During the lockdown, as a result of social distancing, meetings, work, and schooling were set up online. The mean time spent on the use of the Internet and social media has increased significantly. In 2020, US users spent 65 min daily on social media compared to 54 min and 56 min in the preceding years [12]. The above-mentioned data suggest that the development of digital media that took place in recent years has had an impact on all areas of life. In 2002, J. Brown conducted a study in which he noticed that sexual content in mass media may have an impact on beliefs and actual sexual behavior. Back then, people were asking themselves whether people had sex with more partners or without any feelings, because they were seeing this behavior in the media and repeating these patterns [13]. Furthermore, the COVID-19 pandemic may have exacerbated this impact. The aim of this study was to investigate the relationship between social media use and female sexual functioning during the COVID-19 pandemic.

## 2. Materials and Methods

We designed a cross-sectional study using the FSFI (Female Sexual Functioning Index) and BSMAS (Bergen Social Media Addiction Scale) questionnaires and sociodemographic interviews. A self-report questionnaire was distributed on social media. The study group included 546 women aged 16 to 60 years. Data were collected between April and November 2021. Participants reported their sexual behavior, use of social media, and use of computer games, and answered sociodemographic questions. The sociodemographic interviews considered age, place of residence, education, marital status and chronic diseases. In addition, participants were asked about their use of Internet, their most frequently used portals, and their attitude towards pornography. Sexual behavior was investigated using the FSFI questionnaire. The FSFI is a short questionnaire that can be used as a multidimensional self-report tool to assess sexual function in women. The advantages of this questionnaire are its ease of execution and, furthermore, it was designed in such a way that it could assess the sexual function and quality of life of women in clinical and epidemiological studies. The questionnaire consisted of 19 questions about sexual activity within the last 4 weeks. It included six domains, including desire, arousal, lubrication, orgasm, satisfaction, and pain. The maximum score for each domain is 6.0, obtained by summing the responses to the items and multiplying by a correction factor. The domain factor ratio is 0.6 for desire, 0.3 for arousal, 0.3 for lubrication, 0.4 for orgasm, 0.4 satisfaction, and 0.4 for pain [14]. The total composite score for sexual function is a sum of domain scores and ranges from 2.0 (not sexually active and without desire) to 36.0 [15]. In our study, we used the Polish version of the questionnaire, validated by Nowosielski [16]. Social media addiction was measured using the Bergen Social Media Addiction Scale (BSMAS), which is a six-item instrument used to assess the risk of social media disorder (SMD) [17]. The questions relate to the following aspects: salience, mood modification, tolerance, withdrawal, conflict, and relapse [18]. We administered the Polish version of the scale [9]. Women were asked to rate each question on a 5-point Likert scale (1 = very rarely, 5 = very often), and the criterion was then considered endorsed for the analyses presented in this paper. In our study, the cutoff score was 24 points, which was suggested as optimal by Luo et al. [17].

Participation in this study was voluntary and all women were asked to give their informed consent. For our study, we included sexually active women older than 18 years of age. The study was approved by the Commission of Bioethics at Wroclaw Medical University, Wrocław, Poland.

Data were statistically analyzed using Statistica software v. 13.3 (StatSoft, Tulsa, OK, USA). The data were presented as numbers, percentages, and means with standard deviations. The Shapiro–Wilk test was used to analyze the distribution of the data. For comparisons between groups of variables with a normal distribution, Student’s *t*-test for independent variables was used. The differences were interpreted as statistically significant at *p* < 0.05. Cronbach’s α was used to assess the internal consistency of the questionnaire. An α value higher than 0.7 indicates a good internal consistency. This indicator was calculated for the entire questionnaire, covering all questions from the FSFI (19 questions) to the BSMAS (6 questions) sections. The results indicate that the questionnaire had a good overall internal consistency, with a Cronbach’s α of 0.89. Moreover, internal consistency was assessed separately for both parts of the questionnaire. The Cronbach’s α for the FSFI and BSMAS parts was 0.95 and 0.82, respectively.

## 3. Results

We enrolled 546 women in our study. The mean age of our participants was 23.07 ± 4.69 years old (the data were normally distributed, Shapiro–Wilk test value 0.913, *p* = value 0.06). Our respondents were mostly students, in partnership, and residents of big cities. The characteristics of the study group are presented in Table 1.

In the BSMAS questionnaire, 31 women (5.68%) obtained more than 24 points, which could be interpreted as a high risk of addiction to social media. Detailed data are presented in Table 2.

In the FSFI questionnaire, the mean overall score was 26.59. The highest scores were obtained in the lubrication domain, while the lowest scores were obtained in the pain and desire sections. The mean FSFI score was higher among women who did not use dating apps (*p* < 0.003). In addition, we found that the purposes of dating apps use also determined the different FSFI scores. The highest scores were obtained by women who used dating apps to find a sexual partner. Women who used those apps to find love achieved the lowest FSFI scores. The mean FSFI scores were lower among women with a high risk of social media addiction (*p* = 0.01). In addition, we found no statistically significant impact of time spent on social media on FSFI scores. Detailed data are presented in Table 3 and Table 4.

Women who declared that the negative impact on their relationships with partners was caused by the excessive use of social media reached higher scores on BSMAS (*p* < 0.001). The use of computer games was not correlated with the greater risk of social media addiction. Furthermore, the type of computer games played (such as arcade, fictional, war, MMORPG, logical, or simulation) similarly had no impact on higher BSMAS scores. We also found that the use of computer games did not have an impact on FSFI scores. The type of computer game played did not increase the risk of addiction to social media, with one exception: arcane games (*p* = 0.002). Detailed data are presented in Table 5.

The mean FSFI scores were significantly higher in women who declared that they used pornography. However, the frequency of pornography use was not correlated with the mean FSFI scores. Only 1.83% of women answered that they use pornography every day. Almost one third of the respondents declared that they used pornography several times a month (29.12%). A total of 27.66% answered that they used pornography once a year. A comparable percentage of participants denied using pornography (33.52%). The most popular trend in pornography use was a few times a month or a few times a year. Overall, 40.84% of women declared that their attitude to pornography is neutral. A total of 14.84% answered that it is very negative, 15.38 negative, 18.5% positive, and 10.44% very positive. The most popular were porn videos (63%), porn stories (17.58%), porn photos (8.79%), and porn games (1.65%).

## 4. Discussion

Isolation and loneliness during the COVID-19 lockdown encouraged people to use social media more often. This phenomenon may have bright sides, such as it being easier to contact relatives and a reduction in loneliness. In addition, during the lockdown, the Internet and social media gave people opportunities to engage in remote work, online schooling, and business meetings. However, this situation may also have increased the risk of social media addiction and consequently have induced sexual dysfunction.

Our study revealed that addiction to social media is a common problem among Polish women, as 5.68% of women are at high risk of social media addiction. Furthermore, our investigations showed that social media addiction has a negative impact on women’s sexual health.

The study of Rachubińska et al. showed that 27.2% of Polish women were at risk of Internet addiction and 4.8% were addicted. Of these, 16.4% of the women obtained scores that suggested the possibility of addiction to Facebook, and 14.0% achieved scores that indicated this addiction [19]. The influence of the media on sexual behavior was first described in the last century in an educational bulletin in 1981 [20]. Despite the lack of widespread access to the Internet, media use by adolescents and its potential impact on their sexual behavior were analyzed [21]. Interestingly, even then, concerns emerged about the impact of media information on the sexual attitudes and expectations of young adults. This is confirmed by numerous studies, proving that mass media negatively influences the sexual behavior of adolescents [22]. It is not easy to explain the association between social media addiction and sexual functioning. This problem is likely to be multifactorial, with several factors playing an important role. According to Reddy et al., severe depressive syndromes are associated with lower FSFI scores [23]. Moreover, the study conducted by Rachubińska et al. also revealed that severe depressive symptoms and a feeling of loneliness were related to a higher level of Internet addiction among adult women [19]. Furthermore, our findings pointed out that the too-extensive use of social media can have a negative impact on relationships. This observation was also confirmed by Zawada et al., who showed that the higher the level of Facebook addiction was, the lower one’s satisfaction with their relationship status was [24]. In addition, social media has many other negative impacts on every user, from lowering self-esteem to creating new social norms that encourage increasingly risky behaviors [25]. In addition, our findings are similar to those of Alimoradi et al., who also revealed that social media addiction negatively affects women’s sexual functioning [26]. Asekun-Olarinmoye OS. et al., in their study, showed that as many as 75% of respondents use the Internet to view pornography [27]. Additionally, in a study by Carroll JS et al. conducted in the United States among 813 students, 87% of men and 31% of women reported that they themselves looked for pornography on the Internet. The above studies also confirm the results of our study [28]. Our findings showed that pornography use is a common phenomenon among young women. Almost two thirds of our respondents declared that they had used pornography. Mean FSFI scores were significantly higher among women who answered that they used pornography. This could be explained by the fact that pornography use may be correlated with different attitudes to sexual life, and porn users may be more interested in sex. Moreover, in another study, Solano et al. concluded that females admit more often than males to using pornography with a partner to increase sexual stimulation during partnered sexual activity [29].

In a study conducted by Malki et al., most of the women answered that pornography use had no impact on their sexual life, while 13.4% of women declared that pornography had a positive effect on their sexual life [30]. Only 4% of women admitted that pornography had a negative impact on their sexual life. Furthermore, another study showed that there is no correlation between the FSFI score and pornography use in women [31]. Moreover, Berger et al. noted that pornography had no impact on female sexual dysfunction [31]. In another study conducted on this topic, the authors concluded that a higher frequency of pornography use had no impact on partnered sex, but could cause better sexual functioning during masturbation [32].

Another factor investigated in our study was the use of dating apps. In our study, the use of dating apps was found to be negatively correlated with FSFI scores. The women surveyed answered that they mostly used dating apps to find new people and love, so it could be concluded that they did not have a sexual partner. However, there are still stereotypes about the use of dating apps for the sole purpose of having casual sex [33]. Our findings confirm those of Castro et al., who concluded that there is a variety of reasons why people use dating apps, including socializing, looking for relationships, both sexual and romantic, for entertainment, or out of curiosity [34].

Our study showed that digital media affects women’s sexual life. Internet addiction is a serious problem and psychologists’ and psychiatrists’ attention should be drawn to this problem. Sexual education programs in schools should include information about pornography [35]. Lockdown may have increased the feeling of loneliness during the COVID-19 pandemic. As a result, it could lead to social media addiction [19]. Young people should be informed about strategies to cope with loneliness on a daily basis. Social campaigns should inform people about the risk and consequences of social media and Internet addiction. Information on safe Internet use should be distributed among young people.

### Limitations

The main limitation of the study is that Internet users are mainly young people. It explains the mean age of our respondents of 23 years of age; thus, our sample is not representative of the entire population of Polish women. Both scales used, FSFI and BSMAS, are self-reported questionnaires that could lead to biases such as subjective answers and recall bias. Our questionnaire did not include any questions about living with a partner, which could have had an impact on both sexual activity and the use of social media during the COVID-19 pandemic. Our study only considered the impact of internet addiction on the sexual functioning of women during the COVID-19 pandemic, and was not compared to the pre-pandemic state.

## 5. Conclusions

Social media addiction negatively affected women’s sexual functioning during the COVID-19 pandemic. Dating app users obtained lower FSFI scores than other women. Computer games did not have an impact on women’s sexual functioning. Particular attention should be paid to sexuality education in schools—the programs of such activities should include the topic of online pornography. An increasing number of younger social media users do not feel the need to meet other people outside the virtual world. It is important to promote the safe use of the Internet. Social campaigns regarding the risk of addiction to social media would be extremely important.

It would be beneficial to conduct research in other countries to gain a better understanding of this phenomenon. More research is needed in this area.

## Figures and Tables

**Table 1 ijerph-19-08193-t001:** Characteristics of the group (N = 546).

Variable	Result
Age, years, mean ± SD, (95% CI)	23.07 ± 4.69, (22.67; 23.46)
Education	
Primary	29 (5.31%)
Vocational	6 (1.10%)
Secondary	323 (59.16%)
Higher	188 (34.43%)
Employment status	
High school/university student	326 (59.71%)
Student, employed	86 (15.75%)
Employed	107 (19.60%)
Unemployed	15 (2.75%)
On sick leave	11 (2.01%)
Retired	1 (0.18%)
Marital status	
Single	111 (23.27%)
Married	38 7.97%)
In partnership	328 (68.76%)
Place of living	
Rural area	111 (20.33%)
City > 50,000 inhabitants	76 (13.92%)
City from 50,000 to 100,000 inhabitants	44 (8.06%)
City from 100,000 to 250,000 inhabitants	53 (9.71%)
City above 250,000 inhabitants	262 (47.99%)
Comorbid chronic disease	
No	406 (74.36%)
Yes	140 (25.64%)
How many hours do you spend online at work?	
<1 h	171 (31.32%)
1–2 h	107 (19.60%)
2–4 h	121 (22.16%)
4–7 h	97 (17.77%)
>7 h	50 (9.16%)
How many hours do you spend online to relax?	
<1 h	10 (1.83%)
1–2 h	100 (18.32%)
2–4 h	239 (43.77%)
4–7 h	155 (28.39%)
>7 h	42 (7.69%)

**Table 2 ijerph-19-08193-t002:** Results of the Bergen Social Media Addiction Scale.

Question	Score 1		Score 2		Score 3		Score 4		Score 5	
1. Thinking about social media	42	7.69%	57	10.44%	124	22.71%	178	32.60%	145	26.56%
2. Feeling an urge to use social media more	68	12.45%	113	20.70%	126	23.08%	143	26.19%	96	17.58%
3. The use of social media to forget about personal problems	147	26.92%	114	20.88%	116	21.25%	93	17.03%	76	13.92%
4. No success in cutting down on social media use	192	35.16%	147	26.92%	118	21.61%	68	12.45%	21	3.85%
5. Feeling restless if social media using is prohibited	234	42.86%	170	31.14%	104	19.05%	29	5.31%	9	1.65%
6. Too extensive use of social media has a negative impact on job or studies	165	30.22%	136	24.91%	123	22.53%	81	14.84%	41	7.51%

**Table 3 ijerph-19-08193-t003:** Results of the Female Sexual Functioning Index.

Domain	Scores without Multiplying by a Factor		Scores after Multiplying by a Facotr	
	Score, Mean ± SD	Range	Score, Mean ± SD	Range
Desire	6.96 ± 1.89	2–10	4.18 ± 1.13	1.2–6
Arousal	15.29 ± 5.11	0–20	4.59 ± 1.53	0–6
Lubrication	16.24 ± 5.39	0–20	4.87 ± 1.62	0–6
Orgasm	10.86 ± 4.19	0–15	4.34 ± 1.68	0–6
Satisfaction	11.13 ± 4.64	0–15	4.45 ± 1.85	0.8–6
Pain	10.39 ± 4.78	0–15	4.16 ± 1.91	0–6
Overall score	70.87 ± 21.56	2–95	26.59 ± 7.89	2.0–36

**Table 4 ijerph-19-08193-t004:** Female Sexual Functioning Index scores in relation to the use of dating apps and pornography, as well as the Bergen Social Media Addiction Scale score.

Domain	Answer	FSFI		Shapiro–Wilk *p* Value	*t*-Test *p* Value
		Mean ± SD	95% CI		
Do you use dating apps?	Yes (*n* = 80)	24.20 ± 7.76	22.48; 25.92	0.259	0.003
	No (*n* = 466)	27.00 ± 7.85	26.28; 27.71	0.485	
Do you use pornography?	Yes (*n* = 363)	27.21 ± 6.95	26.49; 27.93	0.411	0.009
	No (*n* = 183)	25.35 ± 9.39	23.98; 26.72	0.239	
BSMAS	0–24 scores (*n* = 515)	26.74 ± 7.84	26.06; 27.42	0.202	0.01
	25–29 scores (*n* = 31)	24.07 ± 8.47	20.97; 27.18	0.078	

**Table 5 ijerph-19-08193-t005:** Bergen Social Media Addiction Scale scores in relation to extensive social media use and playing computer games.

Domain	Answer	BSMAS		Shapiro–Wilk *p* Value	*t*-Test *p* Value
		Mean ± SD	95% CI		
In the last year have you used social media so much that it had a negative impact on your relationship with a partner?	Yes (*n* = 226)	17.47 ± 5.16	16.94; 18.01	0.083	<0.001
	No (*n* = 203)	13.65 ± 4.94	12.96; 14.33	0.071	
Do you play computer games?	Yes (*n* = 251)	15.82 ± 5.01	15.19; 16.44	0.138	0.35
	No (*n* = 295)	16.25 ± 5.59	15.61; 16.89	0.122	

## Data Availability

Data are contained within the article.

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
