# Peer review of "Internet Addiction and Polish Women’s Sexual Functioning: The Role of Social Media, Online Pornography, and Game Use during the COVID-19 Pandemic—Online Surveys Based on FSFI and BSMAS Questionnaires"

_ijerph, 2022, doi:10.3390/ijerph19138193_

Round 1
Reviewer 1 Report
Dear Editor and authors,
Firstly, I thank you for my consideration as a reviewer of this manuscript. It is my pleasure to contribute to Women & Health.
The research explores the impact of social media use on female sexual functioning during the COVID-19 pandemic. This work is well justified and organized, with up-to-date citations. Minor issues that they should consider addressing before publication are described:
In the Introduction section, I suggest including some results in support of the impact of digital media on all areas of life (lines 68-69).
In the method section, I recommend indicating the age range of the participants. Then, the authors should include the information about the original and validation version of FSFI and BSMAS. More information and description about the instruments used should be included. The internal consistency reliability should be indicated for the original or validation version and calculated for this sample. Moreover, more information about the socio-demographic questionnaire should be mentioned. Finally, the procedure should be described in more detail.
In the results section, the socio-demographic characteristics could be compared between the groups of participants. Also, the results between the comparison groups should be included in the Tables. I recommend including more information about the organization of the participants in each group for the comparisons.
The discussion section should be reviewed. The authors would discuss the results in the impact of pornography use on sexual function in relation to the differences between the previous studies and their study. More discussion about the results of the dating app should be described. Moreover, I recommend including the lines 180-185 at the beginning of the section.
Finally, I recommend expanding the conclusion of the research on the implications of the results in sex therapy and highlighting the future research.
Reviewer 2 Report
In this paper entitled "Impact of social media addiction on sexual dysfunction in women", the authors describe how internet use, not just social media, correlates with different scores on the Female Sexual Functioning Index scale.
In the review process, we have identified some aspects of the manuscript that could be improved.
1. Title
The title is not representative of the results, because other uses of the internet are assessed and not only social networks. Moreover, the epidemiological study design should be included in the study.
2. Introduction
In the introduction, there is only data about internet use, however there is nothing that connects internet use with sexual dysfunction, or what is the hypothesis that internet use affects the sexual domain.
The references used are not valid as they appear from non-validated secondary sources. The uses of the internet should be referenced by scientific articles.
Objective: The objective does not correspond to the results, since in addition to social networks, the use of pornography and video games, among other aspects, is evaluated.
3. Materials and Methods
The study design must be described in materials and methods.
The major methodological issue with this article is that there is no stated sample size calculation based on the objectives of the study. The authors should indicate, based on the population data, which sample of the register is necessary and sufficient to be able to evaluate the impact of social media use on female disfuntioning.
UNCONTROLLED BIAS
Target population
As described in the results, there is a bias due to the age of the survey participants. This age makes it representative of only a part of the population, so the target population to which the results can be extrapolated should be reconsidered, given the method of dissemination of the surveys and the final responders.
Period
The surveys were collected from April to November, and during this time the measures of confinement were not homogeneous, as well as the general mood as a function of the duration of confinement. The authors should assess how this confounding factor is affected.
Lockdown influence
To assess the influence of lockdown on outcomes, the authors should have data on sexual dysfunction in women, either at an earlier stage or at a later stage. It is not valid with the measure during confinement, because the authors do not know whether these results would be in other periods.
Missing data
It is described that questionnaires with missing data have been deleted, however it does not appear how many questionnaires have been invalidated. We recommend the authors to include a figure with the inclusion of the participants.
Living together
It has not been controlled in people who are in partnership, or in other situations if they are actively living with a partner. This factor may be an important confounding factor in the results.
Statistics used and data presented
It is described that non-parametric statistical tests have been used, however all quantitative data are represented with mean and standard deviation. This is not correct. The authors should express the data as median and IQR in case a normal distribution is not followed.
4. Results
The result description will be in different order. First include the description of the survey participant, second describe the different use of internet and their frequency and lastly the influence with BSMAS and FSFI.
Table 1. Marital status does not add up to 546, therefore there must be missing data or being a mistake, please indicate the missing data or correct it.
Table 2. This table is very uninformative, as there is no indication of the domains or what each of the questions in the BSMAS questionnaire represents. We recommend the authors to modify it to make it more informative.
Table 3. The range of FSFI scale is 2 to 36 as described in M and M section. However, the inferior limit in this table is 1,2 or 0. Please explain it and correct it.
Table 4. This table will be more explicative as a boxplot figure.
Table 5. The domain “In the las year…” does not add up 546, please explain it and correct it. In the domain of use of pornography is surprising that both Yes and No reach more that 24 at BSMAS scale, when only 31 participant has high scoring. Please review it or explain it.
Table 6. This table will be described in text. We recommend the authors delete it, and include the result in the text.
5. Conclusions
Cross-sectional cohort studies are studies to generate hypotheses and not to prove causality. This study, despite all the uncontrolled biases, can generate the hypothesis of causality between internet use and sexual dysfunction, but never prove it. For this reason, the discussion should focus on the description of the data obtained and the support of the causality hypothesis.
6. References
Please remove all references from statitsta.com and replace them with references from scientific articles.
Round 2
Reviewer 1 Report
I suggest the authors review the relationship between the impact of digital media on human sexuality in the Introduction section.
The internal consistency reliability should be calculated for this sample. More information about the procedure should be described in more detail.
In lines 107-109, the authors had indicated comparisons between groups by Student’s t-test; however, there is no description of the distribution of the groups. For this reason, the authors must review the method and results section. In addition, I recommend examining the discussion section. The authors should mention association terms and they should avoid talking in terms of causality.
